# Continual Robot Learning via Language-Guided Skill Acquisition

**Shuo Cheng**[*]
*School of Interactive Computing*
*Georgia Institute of Technology*

**Zhaoyi Li**[*]
*School of Interactive Computing*
*Georgia Institute of Technology*

**Kelin Yu**[*]
*Department of Computer Science*
*University of Maryland, College Park*

**Danfei Xu**
*School of Interactive Computing*
*Georgia Institute of Technology*

**Reviewed on OpenReview:** *https://openreview.net/forum?id=oYRNxxGN9u*

## Abstract

To support daily human tasks, robots need to tackle complex, long-horizon tasks and continuously acquire new skills to handle new problems. Deep Reinforcement Learning (DRL) offers potential for learning fine-grained skills but relies heavily on human-defined rewards and faces challenges with long-horizon goals. Task and Motion Planning (TAMP) are adept at handling long-horizon tasks but often need tailored domain-specific skills, resulting in practical limitations and inefficiencies. To overcome these complementary limitations, we propose LG-SAIL (Language Models Guided Sequential, Adaptive, and Incremental Skill Learning), a framework that leverages Large Language Models (LLMs) to synergistically integrate TAMP and DRL for continuous skill learning in long-horizon tasks. Our framework achieves automatic task decomposition, operator creation, and dense reward generation for efficiently acquiring the desired skills. To facilitate new skill learning, our framework maintains a symbolic skill library and utilizes the existing model from semantic-related skills to warm start the training. LG-SAIL demonstrates superior performance compared to baselines across six challenging simulated task domains across two benchmarks. Furthermore, we demonstrate the ability to reuse learned skills to expedite learning in new task domains, and deploy the system on a physical robot platform. More results on website: https://sites.google.com/view/continuallearning.

## 1 Introduction

For robots to aid in daily human tasks, they need to tackle intricate long-term challenges and adapt to unfamiliar situations. While Deep Reinforcement Learning (DRL) techniques are promising for acquiring fine-grained manipulation skills (Levine et al., 2016; Gu et al., 2017), they require carefully designed reward functions and often fail to achieve long-term goals in complex environments. Conversely, Task and Motion Planning (TAMP) methods (Garrett et al., 2021) are adept at addressing and adapting to long-term tasks due to their robust state and action abstracts. However, their dependence on expertise to construct the planning domain restricts their practical use in real-world scenarios.

---

[*]Equally Contributed

In an effort to overcome the constraints of both TAMP and skill learning, researchers have proposed the concept of learning skill policies within TAMP systems (Cheng & Xu, 2023; McDonald & Hadfield-Menell, 2022; Mandlekar et al., 2023b). While these approaches have shown promise in enabling skill acquisition for long-horizon tasks efficiently, they still rely on expert knowledge to define planning domains (e.g., symbolic operators with conditions and effects) and dense reward functions. These assumptions limit the scalability of skill learning, especially in real-world contexts where robots frequently confront novel challenges where the planning domains cannot be defined beforehand.

In this work, we introduce LG-SAIL (Fig. 1), a continuous learning framework for robot manipulation that integrates large language models (LLMs) with TAMP and DRL. We observe that LLMs, trained on vast web data, excel at commonsense reasoning for task decomposition and skill creation, while TAMP's compact predicate-based state representations provide strong priors that guide and regularize LLM reasoning, improving efficiency and robustness. Leveraging TAMP's abstract states, we structure task decomposition and skill generation to reduce LLM hallucinations. We also define metric functions from TAMP predicates to enable LLMs to generate dense rewards for fast policy learning. To accelerate skill acquisition, trained policies are stored in a skill library, and relevant skills are retrieved via semantic similarity of LLM encoder features to warm-start training. This creates a virtuous cycle between planning and skill learning, supporting efficient continual robot learning.

To summarize, our key contributions include: 1) using LLMs for continual skill learning via automatic task decomposition, skill creation, and dense reward generation; 2) leveraging structured state representations to regularize LLM content generation; 3) improving skill acquisition by maintaining and reusing a skill library; 4) demonstrating through extensive evaluation on six challenging simulated tasks from two benchmarks that our framework outperforms prior methods and supports real-world deployment, paving the way for lifelong robot learning in complex environments.

## 2 Related Work

### 2.1 LLMs for Reward Design

Designing effective reward functions has long been a challenge in reinforcement learning (Sutton, 2018). Recent studies have investigated using LLMs to generate reward functions (Yu et al., 2023; Xie et al., 2023; Zeng et al., 2024; Ma et al., 2023; 2024). While these methods have demonstrated potential in learning short-horizon skills, their scalability to complex, long-horizon tasks remains uncertain. Free-form LLM-based reward generation is prone to hallucinations, often producing syntax errors and irrelevant content. Moreover, approaches such as Eureka (Ma et al., 2023) are time-intensive, requiring multiple rounds of policy training and evaluation to refine the reward functions for each individual skill, making them impractical for long-horizon task learning. Zeng et al. (Zeng et al., 2024) propose a method that iteratively refines reward functions based on feedback from the task learning process. However, this trial-and-error approach can result in delayed convergence and fluctuating performance, hindering consistent improvements. We address these limitations by parameterizing object relationships with metric functions, using LLMs to compose them for efficient, automated dense reward generation.

### 2.2 LLMs for Planning and Decision Making

In recent years, LLMs have been widely explored for robot planning (Huang et al., 2022b;a; Rana et al., 2023; Ahn et al., 2022a; Liu et al., 2023; Joublin et al., 2024) and decision-making (Singh et al., 2023; Liang et al., 2022; Li et al., 2022). These studies demonstrate LLMs' ability to understand complex commands, create task plans, and translate natural language into executable actions. For instance, SayCan (Ahn et al., 2022a) uses LLMs to generate task plans, grounding each skill with learned control policies, while ProgPrompt (Singh et al., 2023) prompts LLMs to generate executable programs that invoke function calls for robot tasks. Researchers have also explored using LLMs to solve PDDL planning problems (Silver et al., 2024), showing that LLMs can act as generalized planners by generating efficient programs for tasks within a domain. This finding informs our approach. Unlike prior methods that generate skill sequences or programs, we use LLMs'

semantic knowledge to construct planning domains and guide skill learning in TAMP, enabling more efficient and robust handling of long-horizon tasks.

## 2.3 TAMP and Learning for TAMP

Task and Motion Planning (TAMP) (Kaelbling & Lozano-Pérez, 2013; Garrett et al., 2021) offers a robust framework for tackling long-horizon tasks in structured environment settings by decomposing complex planning problems into a sequence of symbolic-continuous subtasks, which simplifies the optimization for finding solutions. However, traditional TAMP systems require manually defined planning domains, which depend on extensive expert knowledge. To reduce this burden, recent works (Konidaris et al., 2018; Silver et al., 2022; McDonald & Hadfield-Menell, 2022; Zhao et al., 2024) have integrated learning into TAMP, including efforts to learn low-level skill policies. Mandlekar et al. (Mandlekar et al., 2023b) proposed a TAMP-gated control mechanism that selectively transfers control between a human teleoperator and the robot, using the collected data to train skill policies and improve the TAMP system. LEAGUE (Cheng & Xu, 2023) takes a different approach by leveraging the symbolic interface of task planners to guide RL-based skill learning, creating abstract state spaces that enable skill reuse and improve scalability for long-horizon tasks. Despite these advances, task decomposition still relies on hand-crafted symbolic planners. Besides using symbolic planners for skill planning, researchers have also explored learned planners or heuristics to compose skills Nasiriany et al. (2022b); Zheng et al. (2025); Mishra et al. (2023), which typically rely on task-specific training data, collected either via trial-and-error interaction or expert demonstrations. In this work, we utilize LLMs to integrate TAMP planning with skill learning, enabling robots to tackle long-horizon tasks more efficiently with minimal human input.

# 3 Background and Problem Setting

**Planning domain and skill representations.** We focus on deterministic, fully observed tasks that can be described in PDDL (Fox & Long, 2003), with object-centric states, continuous actions, and a known transition function. Formally, an environment can be characterized by a tuple $\langle \mathcal{O}, \Lambda, \Psi, \Omega, \mathcal{G} \rangle$. Each object entity $o \in \mathcal{O}$ within the environment (e.g., `peg1`), possesses a specific type $\lambda \in \Lambda$ (such as `peg`) and a tuple of $\text{dim}(\lambda)$-dimensional features containing object state information such as pose and size. The environment state $x \in \mathcal{X}$ is a mapping from object entities to features: $x(o) \in \mathbb{R}^{\text{dim}(\text{type}(o))}$. Predicates $\Psi$ describe the relationships among objects in the environment. Each predicate $\psi$ (e.g., `Holding(?object:peg)`) is characterized by a tuple of object types $\langle \lambda_1, ..., \lambda_m \rangle$ and a binary classifier that determines whether the relationship holds: $c_\psi : \mathcal{X} \times \mathcal{O}^m \to \{True, False\}$, where each substitute entity $o_i \in \mathcal{O}$ is restricted to have type $\lambda_i \in \Lambda$. Each binary classifier is constructed through evaluating a set of metric functions $\mathcal{F}_\psi = \{f_i\}$ related to this predicate: $c_\psi \triangleq \bigwedge_{f_i \in \mathcal{F}_\psi} \mathbb{1}[f_i \leq \epsilon_i]$, where each metric function $f_i$ (e.g., `xy_dis(·,·)`) outputs a real number to quantify the relationships among the query objects (e.g., the distance between two object centers projected onto the xy-plane). Evaluating a predicate on the state by substituting corresponding object entities will result in a ground atom (e.g., `Holding(peg1)`). A task goal $g \in \mathcal{G}$ is represented as conjunction over a set of ground atoms (e.g., `In(peg1,hole1)` $\wedge$ `In(peg2,hole2)`), where a symbolic state $x_\Psi$ can be obtained by evaluating a set of predicates $\Psi$ and dropping all negative ground atoms.

Each lifted symbolic operator $\bar{\omega} \in \bar{\Omega}$ is defined by a tuple $\langle \text{Par}, \text{Pre}, \text{Eff}^+, \text{Eff}^- \rangle$, where `Pre` denotes the precondition of the operator, `Eff`$^+$ and `Eff`$^-$ are lifted atoms that describe the expected effects (changes in conditions) upon successful skill execution. `Par` is an ordered parameter list that defines all object types used in `Pre`, `Eff`$^+$, and `Eff`$^-$. An example of `Pick` operator can be defined as:

```
Pick(?object)
  PAR: [?object:peg]
  PRE: {HandEmpty(),OnTable(?object)}
  EFF⁻: {HandEmpty(),OnTable(?object)}
  EFF⁺: {Holding(?object)}
```

A ground skill operator $\underline{\omega}$ substitutes lifted atoms with object instances: $\underline{\omega} = \langle \bar{\omega}, \delta \rangle \triangleq \langle \underline{\text{Pre}}, \underline{\text{Eff}}^-, \underline{\text{Eff}}^+ \rangle$, where $\delta : \Lambda \to \mathcal{O}$. A symbolic task plan is a sequence of ground operators that, when executed successfully, leads to an environment state that satisfies the task goal.

**MDP.** Learning the grounded low-level skills $\pi$ for any symbolic operators $\omega$ can be formulated as a Markov Decision Process (MDP) denoted by $\langle \mathcal{X}, \mathcal{A}, \mathcal{R}(x,a), \mathcal{T}(x'|x,a), p(x^{(0)}), \gamma \rangle$, with continuous state space $\mathcal{X}$,

continuous action space $\mathcal{A}$, reward function $\mathcal{R}$, environment transition model $\mathcal{T}$, distribution of initial states $p(x^{(0)})$, discount factor $\gamma$. The objective for RL training is to maximize the expected total reward $J$ of the policy $\pi(a|x)$ that the agent employs to interact with the environment:

$$J = \mathbb{E}_{x^{(0)}, x^{(1)}, \ldots, x^{(H)} \sim \pi, p(x^{(0)})} \left[ \sum_t \gamma^t \mathcal{R}(x^{(t)}) \right]. \tag{1}$$

**Problem setting.** Our setting is inspired by prior work on operator invention for bi-level planning (Silver et al., 2023; Chitnis et al., 2022). However, we assume only a small set of predefined predicates and metric functions. Given a task goal $g$, we automate operator discovery for decomposing long-horizon tasks and learn primitive skill policies to achieve the resulting subgoals. Each lifted operator $\bar{\omega}$ is associated with a single skill policy $\pi$, shared across all its groundings during execution.

## 4 Method

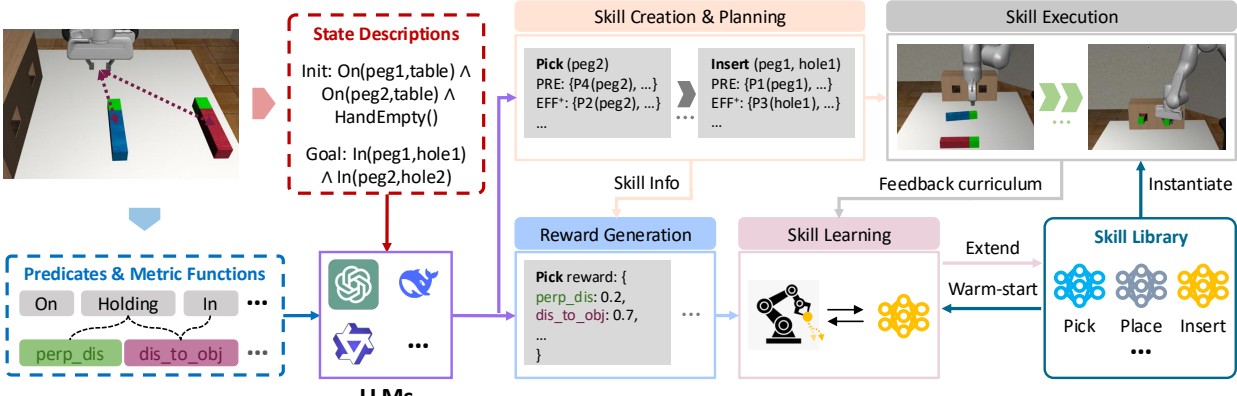

Figure 1: **Framework Overview.** Our framework automates task decomposition, skill creation, and dense reward generation by creating a virtuous cycle between planning and skill learning. Execution failures reveal skills needing improvement, while a skill library enables warm-starting new skills using semantic similarity.

We seek to achieve automatic skill abstraction and continual learning in long-horizon tasks by leveraging the strong priors in LLMs. Our primary contribution is an integrated pipeline (Fig. 1) that uses LLMs for task decomposition and planning (Sec. 4.1), and constructs dense rewards for skill learning (Sec. 4.2). We describe how to accelerate learning of new skills in lifelong learning scenarios in Sec. 4.3.

### 4.1 Task Decomposition and Skill Creation

To reduce the domain knowledge needed for designing valid skills in TAMP, we propose leveraging the rich semantic knowledge of LLMs to decompose tasks into reusable, elementary skills. We use GPT-4 (Achiam et al., 2023) in our experiments. To address the hallucination issue—where generated task plans may overlook constraints between adjacent skills and "hallucinate" impossible effects—we incorporate structural information from TAMP system, including available predicates and the task goal description, as prompts to regularize the outputs. An example of operator generation is shown below:

```
Task: Generate preconditions and effects for the "Pick" action by selecting from the set of
    all available predicates. Ensure the output follows the format [Predicate1, Predicate2,
    ...].
Important: Each predicate is preceded by crucial comments in the code - these comments must
    be retained and considered carefully while generating the preconditions and effects.
- Action: "Pick"
- All predicates: {"Holding", "OnTable", "In", ...}
- The code of those predicates are shown below:
...
```

**Plan verification.** Although the availability of predicates simplifies the skill creation process, the generated operators may occasionally contain irrelevant preconditions and effects. To enhance the accuracy of task decomposition, we validate the generated symbolic operators using the $A^*$ algorithm to produce the symbolic task plan via deterministic search. To plan for a task goal $g$, we first evaluate the predicates on the current environment state $x$, yielding the corresponding symbolic state $x_\Psi$. We then ground each lifted operator $\bar{\omega} \in \bar{\Omega}$ by substituting object entities in the environment in preconditions and effects, leading to ground operators $\underline{\omega} \triangleq \langle \underline{\text{Pre}}, \underline{\text{Eff}}^-, \underline{\text{Eff}}^+ \rangle$ that support operating with symbolic states. A ground operator is considered executable only when its preconditions are satisfied: $\underline{\text{Pre}} \subseteq x_\Psi$. The operators induce an abstract transition model $T(x_\Psi, \underline{\omega})$ that allows planning in symbolic space:

$$x'_\Psi = T(x_\Psi, \underline{\omega}) \triangleq (x_\Psi \setminus \underline{\text{Eff}}^-) \cup \underline{\text{Eff}}^+. \tag{2}$$

Compared to directly using LLMs for task plan generation, incorporating an $A^*$ planner provides feedback by evaluating the proposed operators and verifying whether they are sufficient to reach the task goal. If the search fails, unreachable operators are identified and used to guide LLM-based regeneration, enabling the LLM to produce correct operators with fewer attempts.

## 4.2 Reward Generation and Skill Learning

Once the planning domain is established, and symbolic skills can be planned for specific task goals, we can proceed to learn the corresponding low-level skill policies via deep reinforcement learning. Dense rewards are crucial for reinforcement learning, but human-crafted dense rewards require considerable effort. It presents challenges for long-horizon tasks with various skills and lifelong learning scenarios where agents are exposed to diverse new tasks. We therefore consider using LLMs for automating reward generation.

**Reward construction with metric functions.** LLM-generated reward functions often contain imprecise code or syntax errors (Xie et al., 2023; Ma et al., 2023; Yu et al., 2023), requiring iterations or human feedback. Our key insight is that predicate-derived metric functions in TAMP encode object relationships, providing structural regularization on the generated content. Each metric function $f \in \mathcal{F}$ defines a quantitative relationship between objects: $f : \mathcal{X} \times \mathcal{O}^m \to \mathbb{R}$. For example, `dis_to_obj(robot, peg1)` defines the distance between the robot's gripper and the `peg1` object. Instead of generating code from scratch, we simplify the problem by using LLMs with extracted skills definition from structural skill operators to select metric functions and their corresponding weights to construct dense rewards. This strategy improves the success rate of generating usable reward functions by constraining the LLMs' solution space, thereby reducing irrelevant or erroneous outputs. Additionally, the semantic information from metric functions aids the LLMs in composing reward functions that better align with task objectives.

To better harness the LLMs' proficiency in code interpretation and task comprehension, for each skill operator $\underline{\omega}$, we provide the LLM with the source code of metric functions along with generated skill operators. In this work we use GPT-4 (Achiam et al., 2023). An examplar prompt for reward generation can be found below:

```
Task: Design a reward function for training "Pick" skill using RL based on the template.
- Skill: "Pick"
- Objects: ["peg", ...]
- Metric functions: ["dis_to_obj", ...]
- The template is defined as:
"""
Reward Template:
{'reward_template': [[reward score, 'metric_function(inputs)'], ...]}
Guidelines to complete the reward template:
Select Metric Functions: Choose functions that help in learning the skill (code provided
    below).
Assign Reward Scores: Give each chosen function a score (0.0 to 1.0), with higher scores
    indicating greater importance.
"""
- The code of metric functions are shown below:
...
```

In this example, the metric function `dis_to_obj` with its corresponding code provides semantic information that helps the LLMs understand its purpose is to move the gripper closer to an object. Subsequently, the LLM

generates dense rewards $\mathcal{R}_{\underline{\omega}}^D$ by selecting a subset of relevant metric functions $\mathcal{F}_{\underline{\omega}} = \{f_1, f_2, \ldots, f_n\} \subseteq \mathcal{F}$ and assigning weights $\mathcal{V}_{\underline{\omega}} = \{v_1, v_2, \ldots, v_n\}$ to their normalized forms $f_i^*$, reflecting their relative importance to the reward. The dense reward is thus defined as:

$$\mathcal{R}_{\underline{\omega}}^D(x^{(t)}) = \sum_{i=1}^n v_i \, f_i^*(x^{(t)}, \mathcal{O}_{\underline{\omega}}), \qquad v_i, \, f_i^*(\cdot) \in [0,1] \ \forall i. \tag{3}$$

where $\mathcal{O}_{\underline{\omega}}$ defines a group of object entities relevant to operator $\underline{\omega}$.

To ensure the learned policy achieves the desired outcomes specified by the skill operators, we incorporate sparse rewards derived from predicates. The agent receives a maximal reward when all predicates that define the desired effect of the skill operator $\underline{\omega}$ are met:

$$\mathcal{R}_{\underline{\omega}}^S(x^{(t)}) = \begin{cases} 1 & \text{if } (\bigwedge c_{\underline{\psi^+}}(x^{(t)})) \bigwedge (\bigwedge \neg c_{\underline{\psi^-}}(x^{(t)})) \\ & \quad \text{for } \underline{\psi^+} \in \mathtt{Eff}^+ \text{ for } \underline{\psi^-} \in \mathtt{Eff}^- \\ 0 & \text{otherwise} \end{cases} . \tag{4}$$

This complements the dense rewards constructed by the LLM, promoting the development of more robust skills for long-horizon tasks and ensuring the expected outcomes are met to seamlessly transition to the next skill. Finally, the skill reward is constructed as: $\mathcal{R}_{\underline{\omega}}(x^{(t)}) = \max(\mathcal{R}_{\underline{\omega}}^D(x^{(t)}), \mathcal{R}_{\underline{\omega}}^S(x^{(t)}))$.

**Skill learning with state abstraction.** With the LLM constructed dense rewards, we can train a policy for each symbolic skill using RL. Since the precondition and effect of a ground operator $\underline{\omega}$ induce an effective abstraction of the environment, we can define a skill-relevant state space to prevent the learned policy from being influenced by task-irrelevant objects, thus enhancing its learning efficiency and generalization (Cheng & Xu, 2023): $\hat{x} = \{x(o) : o \in \mathcal{O}_{\underline{\omega}}\}$. The training objective for RL is therefore formulated as:

$$J = \mathbb{E}_{x^{(0)}, x^{(1)}, \ldots, x^{(H)} \sim \pi, p(x^{(0)})} \left[ \sum_t \gamma^t \mathcal{R}_{\underline{\omega}}(x^{(t)}) + \alpha \mathcal{H}(\pi(\cdot | \hat{x}^{(t)})) \right]. \tag{5}$$

We use Soft Actor-Critic (SAC) (Haarnoja et al., 2018) to optimize the skill policy, where $\mathcal{H}$ is the entropy term. Despite the use of dense rewards, RL exploration can be inefficient for learning complex motions (Cheng & Xu, 2023). Conversely, while motion planning from TAMP systems struggles with contact-rich manipulation, it is well-suited for handling free-space motions (Mandlekar et al., 2023b). Building on this, we propose augmenting our policy with motion planner-based transition primitives. The key idea is to first use an off-the-shelf motion planner to approach the skill-relevant object (as specified by the skill operator) before initiating RL-based skill learning. For the motion planning target, we set the goal position $0.04\,\mathrm{m}$ above the object or placement position identified by the task planner. This strategy facilitates more efficient exploration while preserving the ability to learn closed-loop, contact-rich manipulation skills.

### 4.3 Continual Skill Acquisition through Integrated Planning and Skill Learning

So far, we have described how to leverage the rich semantic knowledge from LLMs to guide task decomposition, skill operators creation, and reward generation for continual skill learning. In this section, we describe how to learn skills within the context of a task planning system. This integrated planning and learning approach ensures that the learned skills are compatible with the planner, while continuously expanding the system's capability to solve more tasks.

**Task planning and skill execution.** For any given task goal $g$, we use $A^*$ planner with the generated operators to search a task plan, as described in Sec. 4.1. With the generated task plan, we sequentially invoke the corresponding skill $\pi^*$ to reach the subgoal that complies with the effects of each skill operator $\underline{\omega}$ in the plan. We roll out each skill until it fulfills the effects of the operator or a maximum skill horizon $H$ is reached. To verify whether the $l$-th skill is executed successfully, we obtain the corresponding symbolic state $x_\Psi^l$ by parsing the ending environment state $x^*$. Then, the execution is considered successful only when the environment state $x^*$ conforms to the expected effects: $T(x_\Psi^{l-1}, \underline{\omega}_l) \subseteq x_\Psi^l$. We track failed skills along with

the corresponding initial simulator state $s^*$ to inform the learning curriculum, using this information to reset the environment for focused practice on those failed skills.

**Automated Curriculum.** To efficiently acquire the necessary skills for a multi-step task, we utilize the task planner as an automated curriculum, which enables progressive skill learning. The core idea is to use already proficient skills to achieve the preconditions of those skills which are still required further learning. At a high level, we iteratively alternate between task planning and skill learning until convergence. We track skill failures during executions and apply strict scheduling criteria: a skill is prioritized for further learning (Sec. 4.2) whenever it fails during rollouts. The algorithm is sketched in Alg. 1. Notably, we share the replay buffers across different skill instances (e.g., `Pick(peg1)` and `Pick(peg2)`) that correspond to the same lifted operator. This allows relevant trajectories to be reused, which enhances both learning efficiency and generalization.

**Accelerate learning with skill library.** Efficient learning of complex tasks requires the continual adaptation of acquired skills to new tasks, even across different domains. Our insight is that semantically similar skill operators often share underlying low-level behaviors. For instance, opening a refrigerator and opening a door may involve similar actions and interactions. As a result, policy models from previously learned skills can serve as effective initializations, providing a warm start for learning new skills in novel situations.

Based on this, we propose to maintain a semantic skill library, where each element is a key-value pair. The key represents the semantic embedding of the skill's symbolic definition, which is extracted by a pre-trained LLM encoder (specifically, OpenAI's text-embedding-3-large): $z = \Phi(\omega) \triangleq \Phi(\langle \texttt{Pre}, \texttt{Eff}^-, \texttt{Eff}^+\rangle)$. The value corresponds to the neural network weights of the associated policy.

---

**Algorithm 1** SKILLACQUISITION

---

**input:** `env` (task environment), $g$ (symbolic goal), $\bar{\Psi}$ (state predicates), $\bar{\Omega}$ (operators by LLM), $\mathcal{R}$ (reward functions by LLM), $\Pi$ (skill library), $\Phi$ (LLM encoder)

**start:**
$\mathcal{O}, x^{(0)}, s^{(0)} \leftarrow \texttt{env.get\_state}()$
$x_\Psi^{(0)} \leftarrow \textsc{parse}(x^{(0)}, \mathcal{O}, \bar{\Psi})$                                     ▷ continuous state to symbolic state
$\underline{\Omega} \leftarrow \textsc{ground}(\mathcal{O}, \bar{\Omega})$                                          ▷ get grounded operators
$[\underline{\omega}_1, ..., \underline{\omega}_L] \leftarrow \textsc{search}(x_\Psi^{(0)}, g, \underline{\Omega})$                         ▷ found plan with length L
$t \leftarrow 0$
**while** *Not Converged* **do**
    $\texttt{env.set\_state}(s^{(0)})$
    $[\pi_1^{(t)}, ..., \pi_L^{(t)}] \leftarrow \textsc{retrieve}([\underline{\omega}_1, ..., \underline{\omega}_L], \Pi)$               ▷ warm-start initialization
    $\mathcal{D} \leftarrow \textsc{rollout}(\texttt{env}, \underline{\Omega}, [\pi_1^{(t)}, ..., \pi_L^{(t)}])$         ▷ failed skills and simulator states
    **if** $\mathcal{D} = \emptyset$ **then**
        **break**
    **end if**
    **for** $s, \underline{\omega} \leftarrow \mathcal{D}$ **do**
        $\texttt{env.set\_state}(s)$
        $\pi_i^{(t)} \leftarrow \Pi[\Phi(\underline{\omega})]$
        $\pi_i^{(t+1)} \leftarrow \textsc{SAC}(\texttt{env}, \pi_i^{(t)}, \underline{\omega}, \mathcal{R}_{\underline{\omega}})$                            ▷ RL training
        $\Pi \leftarrow \Pi \cup \{\Phi(\underline{\omega}) : \pi_i^{(t+1)}\}$
    **end for**
    $t \leftarrow t + 1$
**end while**
**return** $\Pi$

---

To leverage the models stored in this library, for any new skill that needs to be acquired, we first extract its feature embedding $z'$. We then identify the most similar skill in the library using cosine similarity and use the corresponding weights to initialize the new skill, which provides a warm start for continual training. Each newly learned skill is subsequently added to the library, expanding it to speed up the learning process.

# 5 Experiment

In this section, we show that LG-SAIL autonomously learns long-horizon tasks and outperforms prior state-of-the-art methods. We demonstrate its ability to adapt skills to novel tasks and domains—crucial for continual learning—and validate our planning and reward designs via ablations. Finally, we deploy the system on a physical robot platform and present both quantitative and qualitative results in the real world.

## 5.1 Experimental Setup

We conduct experiments in six simulated domains. The domain "`StackAtTarget`", "`StowHammer`", "`PegInHole`", and "`MakeCoffee`" are from LEAGUE (Cheng & Xu, 2023), which involve long-horizon reasoning and contact-rich manipulation. The task setups, predicates, and our generated operators and task plan are shown in Fig. 2. To further evaluate the framework's ability to efficiently adapt learned skills to novel tasks, we include the LIBERO benchmark (Liu et al., 2024b), a benchmark in lifelong robot learning that features diverse objects and long-horizon tasks. We test our framework on both "`LIBERO-Spatial`" and "`LIBERO-Object`". The "`LIBERO-Spatial`" set comprises 10 tasks with varying scene configurations, where the robot must pick up a target bowl and place it on the goal plate. The "`LIBERO-Object`" set includes 10 tasks involving different objects with diverse shapes, which requires the robot to grasp the target object and place it in a basket. The setups and our generated operators are shown in Fig. 3.

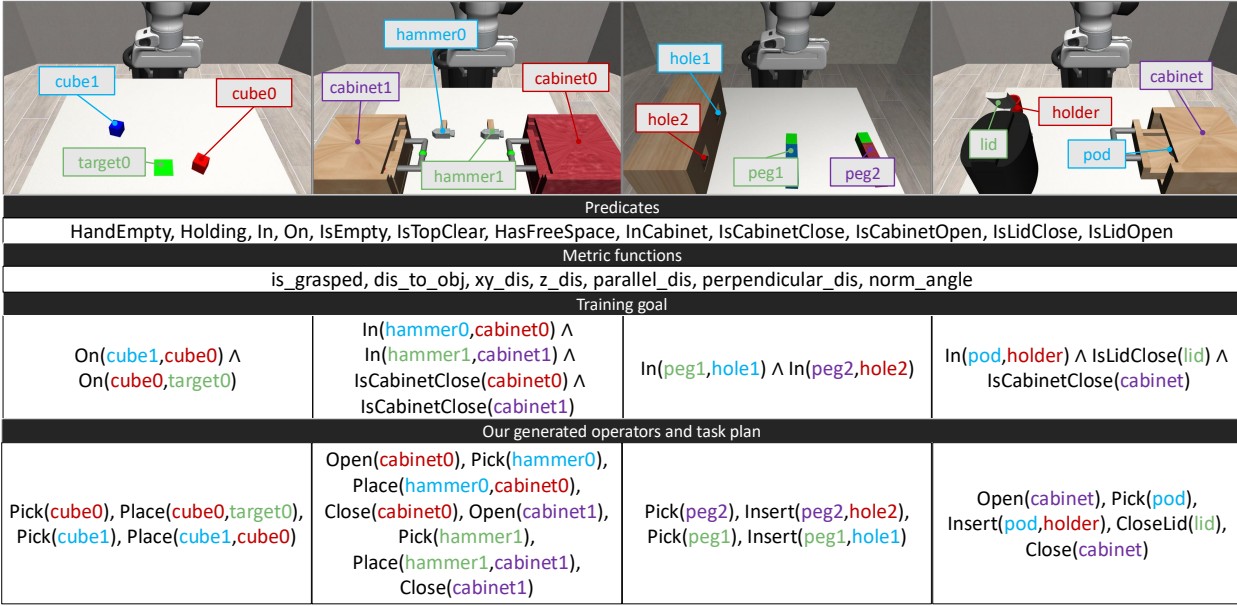

Figure 2: **LEAGUE Tasks (Cheng & Xu, 2023).** Illustration of task setups for "`StackAtTarget`", "`StowHammer`", "`PegInHole`", and "`MakeCoffee`".

All environments are built on the MuJoCo engine (Todorov et al., 2012). We use a Franka robotic arm, controlled at 20Hz with an operational space controller (OSC), providing 5 degrees of freedom: end-effector position, yaw angle, and gripper position.

## 5.2 Quantitative Evaluation

To assess performance, we compare LG-SAIL against strong baselines for learning long-horizon tasks in "`StackAtTarget`", "`StowHammer`", and "`PegInHole`":

- **RL (SAC)** - We utilize Soft Actor-Critic (SAC) (Haarnoja et al., 2018) for vanilla RL baseline;
- **Curriculum RL (CRL)** - This baseline follows the same curriculum strategy as prior works (Sharma et al., 2021; Uchendu et al., 2023), starting training from near-success states and gradually shifting the reset states back to the initial task states. We use the same model as **RL (SAC)** for fair comparison;

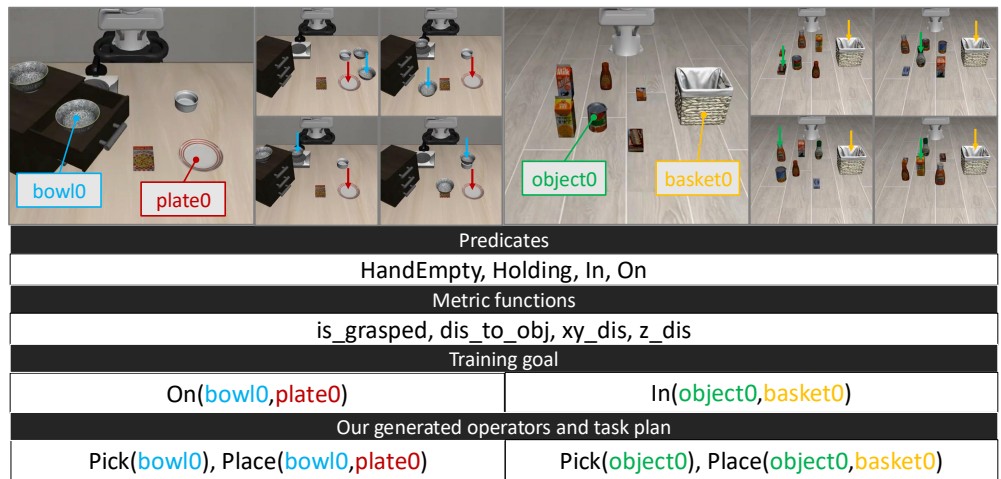

Figure 3: **LIBERO (Liu et al., 2024b).** Tasks setup for "LIBERO-Object" and "LIBERO-Spatial".

- **Hierarchical RL (HRL)** - This baseline employs HRL designs (Dalal et al., 2021; Nasiriany et al., 2022a), where a high-level controller composes parameterized skill primitives. Our implementation builds on MAPLE (Nasiriany et al., 2022b) and uses the oracle task plan to identify target objects for affordance definition.

- **LEAGUE** (Cheng & Xu, 2023) - This baseline reflects the recent trend of integrating TAMP with skill learning. We utilize it for this experiment due to its superior performance on long-horizon tasks.

To ensure fair comparison, we follow experiment settings in LEAGUE (Cheng & Xu, 2023), and we adopt task progress as evaluation metric, which is defined as the summed reward of all task stages and normalized to [0, 1]. The results are presented in Fig. 4.

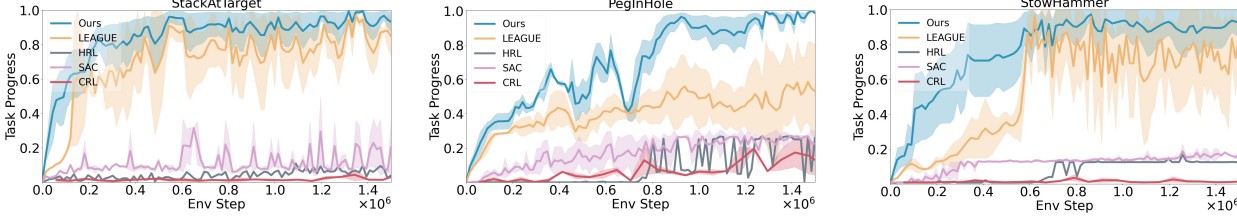

Figure 4: **Baseline comparison.** We compare our framework with other baselines across three task domains. The plot illustrates the average task progress during evaluation over the training phase, measured as the sum of rewards for each successfully executed skill in the task plan, normalized to 1. The shaded area represents the standard deviation for 5 random seeds.

**Our framework efficiently and autonomously learns long-horizon manipulation tasks.** By just providing predicates, metric functions, and task goals of different domains (shown in Fig. 2), our framework autonomously achieves task decomposition, skill creation, reward generation, and integrates TAMP with skill learning. Additionally, we found that methods incorporating skill abstraction and planning (i.e., LG-SAIL and LEAGUE (Cheng & Xu, 2023)) significantly outperform other baselines, underscoring the importance of explicit skill reuse in multi-step tasks with repeating structures. Notably, LG-SAIL demonstrates slightly higher learning efficiency compared to LEAGUE (Cheng & Xu, 2023), potentially reflecting the advantages of LLM-generated rewards over handcrafted ones, as supported by recent literature (Ma et al., 2023; 2024).

### 5.3 Validating Skill Generalization and Adaptation

We design experiments to evaluate the system's continual learning capability by constructing tasks that require varying levels of adaptation.

**Generalizing to new goals.** In addition to training goals for "`PegInHole`" and "`StowHammer`", we directly evaluate our framework on new goals without further learning. For "`StowHammer`", the first test goal is to swap the hammer-cabinet mapping, while the second goal is to place `hammer1` into `cabinet0` while keeping `cabinet1` open. For "`PegInHole`", the first test goal is to swap the peg-hole mapping, and the second goal is to insert `peg1` into `hole2`. The results are presented in Tab. 1.

Table 1: **Task goal generalization.** We evaluate RL (SAC) and LG-SAIL on novel task goals in the "`StowHammer`" and "`PegInHole`" domains under zero-shot settings.

| Domain | Method | Training Goal | Test Goal 1 | Test Goal 2 |
|---|---|---|---|---|
| "`StowHammer`" | RL (SAC) | $0.15 \pm 0.02$ | $0.07 \pm 0.02$ | $0.09 \pm 0.01$ |
| | LG-SAIL | $\mathbf{0.96 \pm 0.15}$ | $\mathbf{0.91 \pm 0.09}$ | $\mathbf{0.71 \pm 0.28}$ |
| "`PegInHole`" | RL (SAC) | $0.19 \pm 0.03$ | $0.09 \pm 0.01$ | $0.10 \pm 0.01$ |
| | LG-SAIL | $\mathbf{0.92 \pm 0.11}$ | $\mathbf{0.59 \pm 0.14}$ | $\mathbf{0.93 \pm 0.06}$ |

We observe that LG-SAIL exhibits minimal performance drop when generalizing to new goals within the domain without additional training, demonstrating strong compositional generalization capabilities.

**Adapting to new objects and scenes.** We aim to evaluate LG-SAIL's ability to adapt learned skills to novel situations using the LIBERO (Liu et al., 2024b) benchmark. The "`LIBERO-Object`" set requires the robot to pick up various target objects and place them into a basket, testing its ability to adapt to new object shapes and poses. The "`LIBERO-Spatial`" set features diverse scene layouts where the robot must place a target bowl on a plate, emphasizing generalization to different object poses and scene configurations. We sequentially test LG-SAIL on tasks from both "`LIBERO-Object`" and "`LIBERO-Spatial`", with results shown in Fig. 5. Interestingly, while LG-SAIL takes some time to learn the initial task, as more skills are accumulated, the learning of subsequent tasks accelerates significantly, demonstrating its strong potential for continual and lifelong learning.

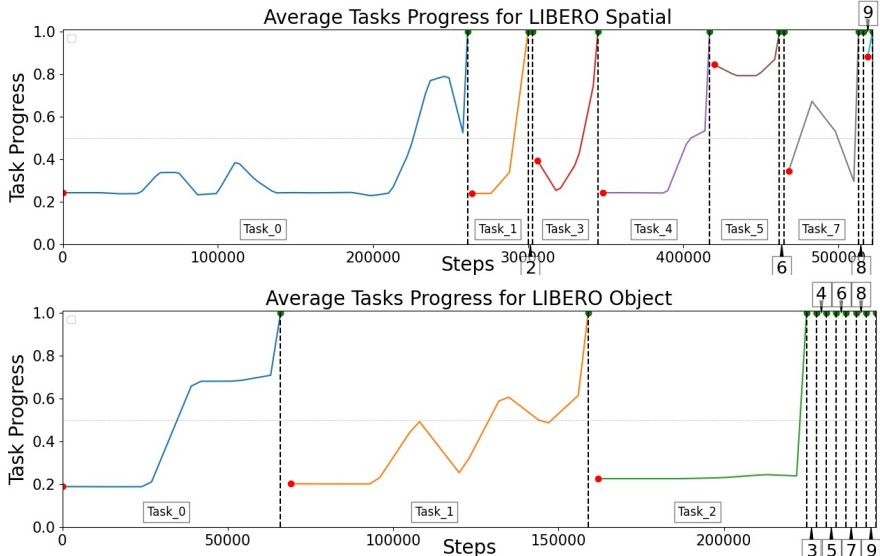

Figure 5: **Continuous skill adaptation and learning.** The learning process is shown for both "`LIBERO-Object`" and "`LIBERO-Spatial`" tasks.

## 5.4 Ablation Study

We evaluate various design choices for task planning and reward generation in LG-SAIL.

**Task planning.** We compare three design choices of using LLMs for skill generation and planning:

**Learning in new domains.** Another advantage of LG-SAIL is its ability to reuse learned models from other domains to warm-start the learning of similar skills in new domains.

To validate this, we evaluate LG-SAIL on "MakeCoffee". LG-SAIL retrieves skills `Open`, `Close`, `Pick`, and `Place` from "`StowHammer`" in the skill library using semantic embeddings and uses these models to initialize skill policies, leading to improved learning efficiency compared to training from scratch (Fig. 6). Compared to `League`, LG-SAIL can retrieve skills that are similar but not identical by measuring semantic distances over symbolic operators—such as warm-starting the novel skill `Insert` with `Place` from "`StowHammer`"—a transfer that is difficult to specify manually. In addition, this task again demonstrates that LLM-generated rewards provide more informative supervision than human-designed rewards.

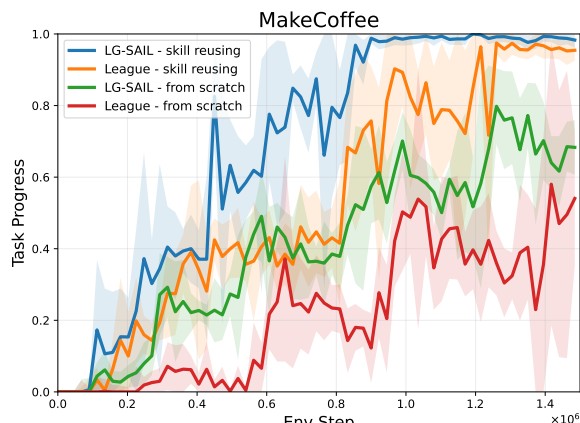

Figure 6: **Skill adaptation in new domains.** For "`MakeCoffee`", we compare learning from scratch and adapting skills learned in the "`StowHammer`" domain with both LG-SAIL and `League`.

- A* Planner (**LLM+A***): This variant is adopted in LG-SAIL, where LLMs are used for skill generation, and A* search is utilized to find task plan based on the generated operators. If no valid task plan is found, the skill generation process is re-invoked to refine the operators.

- LLM Planner (**LLM**): Similar to prior works on LLM-based planning (Ding et al., 2023; Kannan et al., 2024), this variant uses LLMs for both skill operator generation and task planning. If the generated plan is not executable, the process is re-invoked to produce a new plan.

- LLM Planner w/o Replanning (**LLM 1-shot**): This variant skips task plan validation and invokes skill generation and task planning only once.

We evaluate "`StowHammer`" across four aspects: (1) success rate of generating skill operators with correct preconditions and effects for connecting other skills, (2) success rate of generating valid plans conditioned on correctly generated skills, (3) overall task success rate per method (for **LLM+A*** and **LLM**, skill generation is retried until a valid plan is found), and (4) average number of attempts. Each variant runs until 25 correct plans are generated.

Results are in Tab. 2. We found that both design choices with a replanning strategy achieved 100% success rate, while the **LLM 1-shot** variant only reached a 36% success rate. Additionally, the **LLM + A*** approach guarantees a 100% success rate if correct skills are generated, effectively reducing LLM-based planning hallucinations and minimizing the number of replanning attempts.

Table 2: **Task Planning Comparison.** We evaluate the performance of different designs on "`StowHammer`". The task planning success rate is calculated based on trials where the skills are correctly generated.

|  | LLM+A* | LLM | LLM 1-shot |
|---|---|---|---|
| Skill Generation Success Rate (↑) | **61.0%** | 59.3% | 52.0% |
| Task Planning Success Rate (↑) | **100.0%** | 78.1% | 69.2% |
| Full Design Success Rate (↑) | **100.0%** | **100.0%** | 36.0% |
| Average Attempts (↓) | **1.64** | 2.16 | N/A |

**Reward generation.** We validate the impact of full metric functions on reward generation by comparing our approach (**Full MF**) to an ablated variant (**W/O MF**) that provides only metric function's header to the LLM, omitting both the code implementation and descriptive comments. Both are tested on "`StowHammer`" over 25 trials, evaluating reward validity for each skill. Following Xie et al. (Xie et al., 2023), we classify errors into four types: (1) *Class attribute misuse* - incorrect object selection for metric functions, (2) *Attribute*

*hallucination* - referencing non-existent entities, (3) *Syntax/format errors* - structural mistakes, and (4) *Metric selection errors* - inappropriate function choices for rewards. Success is based on error-free reward generation, not its impact on policy optimization.

Table 3: **Reward generation comparison.** We compare reward generation success rates for different skills and the full task on "`StowHammer`".

|         | Full Task | `Pick` | `Place` | `Open`  | `Close` |
|---------|-----------|--------|---------|---------|---------|
| W/O MF  | 20.0%     | 40.0%  | 36.0%   | 44.0%   | 48.0%   |
| Full MF | **92.0%** | **96.0%** | **96.0%** | **100.0%** | **100.0%** |

Tab. 3 presents the reward generation success rates for "`StowHammer`". Our method achieves 92% overall success, with 96% for `Pick` and `Place`, and 100% for `Open` and `Close`. The low error rate underscores its reliability for long-horizon tasks and potential for lifelong learning.

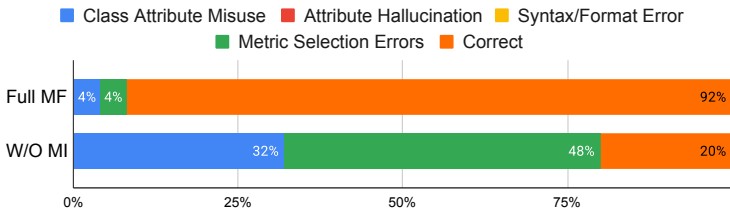

Figure 7: **Error breakdown.** We show the error distribution of reward generation for "`StowHammer`".

Fig. 7 illustrates the error distribution across reward generation strategies. In LG-SAIL, using improved prompts and detailed metric functions reduces attribute/object selection errors from 80% to 8%. This emphasizes that structured information, like quantified object relationships, enhances LLMs' scene interpretation and reward accuracy.

### 5.5 Real World Results

We demonstrate the transfer of our simulation-trained LG-SAIL system to two real-world tasks: "`StackAtTarget`", where the robot stacks two cubes in a target region in a specified order, and "`SortBowl`", which is the task in LIBERO Benchmark that involves placing a bowl on a target plate.

Our system employs a Franka Emika robotic arm and a Microsoft Azure Kinect camera for RGBD image capture. We use AprilTag (Olson, 2011) to detect the 6D poses of relevant objects and perform state estimation to synchronize with the simulated environment. Skills generated by LG-SAIL in simulation are executed in the real world through open-loop control.

We conduct 10 trials per task, with key execution frames in Fig. 8. While occasional failures occur due to occlusion-induced pose estimation errors and bowl slippage from insecure grasps, we achieve overall success rates of 0.8 and 1.0, highlighting the robustness of our system.

## 6 Conclusions and Limitations

We present LG-SAIL, a framework that integrates LLM-guided skill generation and learning with TAMP, enhancing automatic and continuous robot learning across various tasks. Our results show that LG-SAIL outperforms previous approaches by reducing human intervention and improving learning efficiency.

Despite strong performance on challenging long-horizon tasks, we observe common failure modes, including object slippage from unsecured grasps and unexpected contact behaviors. These issues could be mitigated by incorporating failure context into reward generation and by improving robustness during execution. While LG-SAIL currently assumes accurate object states and hand-crafted predicates, recent advances in visual perception (Wen et al., 2024), object relation modeling (Yuan et al., 2022), and predicate learning (Migimatsu & Bohg, 2022; Silver et al., 2023) offer promising paths toward real-world deployment.

Goal: On(bowl0, plate0)

Goal: OnTarget(cube1, target0) ∧ On(cube0, cube1)

Figure 8: **Real-world results.** We demonstrate the deployment of LG-SAIL for real-world tasks.

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

## A  Metric Function Definitions

Table 4: **Metric Functions.** We list all metric functions available in the task environments and provide their definitions.

| Metric Function | Definition |
|---|---|
| is_grasped | Indicates whether the object is currently being held by the gripper |
| dis_to_obj | Distance between gripper and query object |
| xy_dis | 2D Euclidean distance between gripper and object on the XY plane |
| z_dis | Vertical (Z-axis) distance between gripper and object |
| parallel_dis | 2D Euclidean distance between two objects on the XY plane |
| perpendicular_dis | Vertical (Z-axis) distance between two objects |
| norm_angle | Angle difference between two objects' orientations |

Tab. 4 summarizes the metric functions used in our tasks, along with their corresponding definitions.

## B  Task Decomposition and Skill Operator Generation

### B.1  Prompt Examples for Task Decomposition

We provide the complete prompt used for the "`PegInHole`" task as an example for generating required actions:

```
Task: Decompose the given task into a sequence of reusable parameterized skills that can
    achieve the specified end goals from the start states. Output should be in the format: [
    skill1(?object_type), skill2(?object_type1, ?object_type2), ...]

Important:
Skills should use type-parameterized variables (e.g., ?peg, ?hole) rather than specific
    instances (e.g., peg1, hole1)
The skill sequence should represent a general solution pattern that can be applied to any
    valid object instances
Skills should be composable and reusable across similar tasks

Environment:
"
(:init
    (HandEmpty)
    (OnTable peg1)
    (OnTable peg2)
    (IsClear hole1)
    (IsClear hole2)
)
(:goal
    (and
        (In peg1 hole1)
        (In peg2 hole2)
    )
)
"

Available Predicates:
{"Holding(?object)", "OnTable(?object)", "In(?peg, ?hole)", "IsPeg(?object)", "IsHole(?
    object)", "IsClear(?hole)", "HandEmpty()"}

Output Requirements:
Generate a minimal set of parameterized skills that represent the task decomposition
Use only type variables (e.g., ?peg, ?hole) in the skill parameters, NOT specific instances
Skills should capture the abstract manipulation pattern, independent of how many times it
    needs to be repeated
Output ONLY the skill sequence array - no explanations, no extra text
Format: [Skill1(?type1), Skill2(?type1, ?type2), ...]
```

The output actions:

```
[Pick(?peg), Insert(?peg, ?hole)]
```

After task decomposition, the next step is generating symbolic operators for each action, and then using A$^*$ for planning.

## B.2 Prompt Examples for Operator Generation

We provide the complete prompt used for the "`PegInHole`" task as an example:

```
Task: Generate the preconditions (PRE) and effects (EFF) for the action "Pick" using only
    the given set of predicates.
The output must strictly follow the exact format below, with no additional text or
    explanations:

Pick(?peg)
PAR: [?peg:peg]
PRE: {<list of preconditions>}
EFF-: {<list of negative effects>}
EFF+: {<list of positive effects>}

Important:
1. All possible predicates are: {"Holding(?object)", "OnTable(?object)", "In(?peg, ?hole)",
    "IsPeg(?object)", "IsHole(?object)", "IsClear(?hole)", "HandEmpty()"}
2. The definitions of each predicate are provided below.
   These predicates definitions MUST be considered when deciding if the predicate should be
    used in PRE or EFF.
   Do not invent new predicates.
3. All variable names must exactly match the placeholders shown in the format (e.g., ?object
    ).
4. PRE is the set of conditions that must be true before the action can be executed.
5. EFF- is the set of conditions that will no longer hold after the action.
6. EFF+ is the set of conditions that will become true after the action.
7. You must keep the curly-brace {} notation exactly as shown, with predicates separated by
    commas.
8. Only include logically correct predicates based on the given code comments and the
    meaning of "Pick".

- Action: "Pick"
- All predicates: {"Holding(?object)", "OnTable(?object)", "In(?peg, ?hole)", "IsPeg(?object
    )", "IsHole(?object)", "IsClear(?hole)", "HandEmpty()"}
- Predicate definitions:
  def Holding(state, obj):
    # The peg is grasped by the gripper
    return is_grasped(obj)

  def In(state, peg, hole):
    # The peg is in the hole
    d = parallel_dis(state, peg, hole)
    t = perpendicular_dis(state, peg, hole)
    cos = norm_angle(state, peg, hole)
    return d < 0.05 and -0.05 <= t <= 0.05 and cos > 0.95

  ... (other predicates definitions here)

Task: Generate the preconditions (PRE) and effects (EFF) for the action "Insert" using only
    the given set of predicates.
The output must strictly follow the exact format below, with no additional text or
    explanations:

Insert(?peg, ?hole)
PAR: [?peg:peg, ?hole:hole]
PRE: {<list of preconditions>}
EFF-: {<list of negative effects>}
EFF+: {<list of positive effects>}

Important:
1. All possible predicates are: {"Holding(?object)", "OnTable(?object)", "In(?peg, ?hole)",
    "IsPeg(?object)", "IsHole(?object)", "IsClear(?hole)", "HandEmpty()"}
2. The definitions of each predicate are provided below.
   These predicates definitions MUST be considered when deciding if the predicate should be
    used in PRE or EFF.
   Do not invent new predicates.
3. All variable names must exactly match the placeholders shown in the format (e.g., ?object
    ).
4. PRE is the set of conditions that must be true before the action can be executed.
5. EFF- is the set of conditions that will no longer hold after the action.
```

```
6. EFF+ is the set of conditions that will become true after the action.
7. You must keep the curly-brace {} notation exactly as shown, with predicates separated by
   commas.
8. Only include logically correct predicates based on the given code comments and the
   meaning of "Insert".

- Action: "Insert"
- All predicates: {"Holding(?object)", "OnTable(?object)", "In(?peg, ?hole)", "IsPeg(?object
  )", "IsHole(?object)", "IsClear(?hole)", "HandEmpty()"}
- Predicate definitions:
  def Holding(state, obj):
    # The peg is grasped by the gripper
    return is_grasped(obj)

  def In(state, peg, hole):
    # The peg is in the hole
    d = parallel_dis(state, peg, hole)
    t = perpendicular_dis(state, peg, hole)
    cos = norm_angle(state, peg, hole)
    return d < 0.05 and -0.05 <= t <= 0.05 and cos > 0.95

  ... (other predicates definitions here)
```

For the `Holding` predicate, the classifier uses the metric function `is_grasped` to evaluate whether the query object `obj` is held by the robot. For the `In` predicate, the corresponding classifier uses `parallel_dis`, `perpendicular_dis`, and `norm_angle` to determine whether the peg object `obj1` is inserted into the hole object `obj2`.

### B.3 Generated Skill Operators

Below are examples of some skill operators generated by our framework for LEAGUE tasks.

"PegInHole":

```
Pick(?peg)
  PAR: [?peg:peg]
  PRE: {HandEmpty(), OnTable(?peg)}
  EFF⁻: {HandEmpty(), OnTable(?peg)}
  EFF⁺: {Holding(?peg)}

Insert(?peg, ?hole)
  PAR: [?peg:peg, ?hole:hole]
  PRE: {Holding(?peg), IsClear(?hole)}
  EFF⁻: {Holding(?peg), IsClear(?hole)}
  EFF⁺: {In(?peg, ?hole)}
```

"StowHammer":

```
Pick(?object)
  PAR: [?object:object]
  PRE: {HandEmpty(), OnTable(?object)}
  EFF⁻: {HandEmpty(), OnTable(?object)}
  EFF⁺: {Holding(?object)}

Place(?object, ?cabinet)
  PAR: [?object:object, ?cabinet:cabinet]
  PRE: {Holding(?object), IsEmpty(?cabinet), IsCabinetOpen(?cabinet)}
  EFF⁻: {Holding(?object), IsEmpty(?cabinet)}
  EFF⁺: {In(?object, ?cabinet), HandEmpty()}

Open(?cabinet)
  PAR: [?cabinet:cabinet]
  PRE: {HasFreeSpace(), IsCabinetClosed(?cabinet), HandEmpty()}
  EFF⁻: {HasFreeSpace(), IsCabinetClosed(?cabinet)}
  EFF⁺: {IsCabinetOpen(?cabinet)}

Close(?cabinet)
  PAR: [?cabinet:cabinet]
  PRE: {IsCabinetOpen(?cabinet), HandEmpty()}
```

```
EFF⁻: {IsCabinetOpen(?cabinet)}
EFF⁺: {HasFreeSpace(), IsCabinetClosed(?cabinet)}
```

## C   Reward Generation

### C.1   Prompt Examples

As an example, we include the prompt used to generate skill rewards for the "`PegInHole`" task:

```
Task: Generate a reward function dictionary for training the "Pick" skill in RL.

Skill:
Pick(?peg)
  PAR: [?peg:peg]
  PRE: {HandEmpty(), OnTable(?peg)}
  EFF-: {HandEmpty(), OnTable(?peg)}
  EFF+: {Holding(?peg)}

Objects: ["peg"]
Available Normalized Metric Functions: ["dis_to_obj", "orientation_diff", "parallel_dis", "
    perpendicular_dis"]

Reward Template format:
{
    "<metric_function>": <reward_score>,
    ...
}

Important Rules:
1. Output must be exactly one JSON dictionary with keys selected only from the given Metric
    Functions.
2. The definitions of each normalized metric function are provided below.
   These metric function definitions MUST be considered when deciding whether a metric
    function should be used in the composed reward and its corresponding reward score.
   Do not invent new metric functions.
3. The sum of all reward scores will usually be less than 1.0, since a total reward of 1.0
    represents full task success. The purpose of this reward dictionary is to provide a
    smooth shaping signal, allowing the reward to gradually increase as the agent approaches
     success.
4. Each keys value must be a float between 0.0 and 1.0 (inclusive).
5. No additional text, no explanation, no comments, no formatting other than the JSON
    dictionary.
6. The chosen metric functions must logically contribute to learning the "Pick" skill based
    on the given function definitions.
7. The only allowed output format is:
{
    "<metric_function>": <reward_score>,
    ...
}
8. Do not include any metric function not listed in the available set.
9. Absolutely no text outside the dictionary.

- The code of normalized metric functions are shown below:

def dis_to_obj(self, state, obj):
  # Calculate the distance from the robot gripper to the object, normalized it to [0, 1]
  tar_obj_state = state[obj]
  grip_dist = np.linalg.norm(tar_obj_state["grip_dist"])
  r_reach = (1 - np.tanh(10.0 * grip_dist))
  return r_reach

def parallel_dis(self, state, peg, hole):
  # Calculate the reward for aligning the peg with the hole, normalized it to [0, 1]
  peg_state = state[peg]
  hole_state = state[hole]
  peg_pos = peg_state["pos"]
  hole_pos = hole_state["pos"]
  hole_mat = hole_state["mat"]
  hole_mat.shape = (3, 3)
  center = hole_pos # hole_mat @ np.array([0.1, 0, 0])
  hole_normal = hole_mat @ np.array([1, 0, 0]) # np.array([0, 0, 1])
  t = (hole_pos - peg_pos) @ hole_normal
  t_tanh_mult = 4.0 # DEFAULT_PEG_IN_HOLE_CONFIG
```

```
  t_rew = 1 - np.tanh(t_tanh_mult * np.abs(t))
  return t_rew

... (other normalized metric functions definitions here)
```

```
Task: Generate a reward function dictionary for training the "Insert" skill in RL.

Skill:
Insert(?peg, ?hole)
  PAR: [?peg:peg, ?hole:hole]
  PRE: {Holding(?peg), IsClear(?hole)}
  EFF-: {Holding(?peg), IsClear(?hole)}
  EFF+: {In(?peg, ?hole)}

Objects: ["peg", "hole"]
Available Normalized Metric Functions: ["dis_to_obj", "orientation_diff", "parallel_dis", "
    perpendicular_dis"]

Reward Template format:
{
    "<metric_function>": <reward_score>,
    ...
}

Important Rules:
1. Output must be exactly one JSON dictionary with keys selected only from the given Metric
    Functions.
2. The definitions of each normalized metric function are provided below.
   These metric function definitions MUST be considered when deciding whether a metric
    function should be used in the composed reward and its corresponding reward score.
   Do not invent new metric functions.
3. The sum of all reward scores will usually be less than 1.0, since a total reward of 1.0
    represents full task success. The purpose of this reward dictionary is to provide a
    smooth shaping signal, allowing the reward to gradually increase as the agent approaches
     success.
4. Each keys value must be a float between 0.0 and 1.0 (inclusive).
5. No additional text, no explanation, no comments, no formatting other than the JSON
    dictionary.
6. The chosen metric functions must logically contribute to learning the "Insert" skill
    based on the given function definitions.
7. The only allowed output format is:
{
    "<metric_function>": <reward_score>,
    ...
}
8. Do not include any metric function not listed in the available set.
9. Absolutely no text outside the dictionary.

- The code of normalized metric functions are shown below:

def dis_to_obj(self, state, obj):
  # Calculate the distance from the robot gripper to the object, normalized it to [0, 1]
  tar_obj_state = state[obj]
  grip_dist = np.linalg.norm(tar_obj_state["grip_dist"])
  r_reach = (1 - np.tanh(10.0 * grip_dist))
  return r_reach

def parallel_dis(self, state, peg, hole):
  # Calculate the reward for aligning the peg with the hole, normalized it to [0, 1]
  peg_state = state[peg]
  hole_state = state[hole]
  peg_pos = peg_state["pos"]
  hole_pos = hole_state["pos"]
  hole_mat = hole_state["mat"]
  hole_mat.shape = (3, 3)
  center = hole_pos # hole_mat @ np.array([0.1, 0, 0])
  hole_normal = hole_mat @ np.array([1, 0, 0]) # np.array([0, 0, 1])
  t = (hole_pos - peg_pos) @ hole_normal
  t_tanh_mult = 4.0 # DEFAULT_PEG_IN_HOLE_CONFIG
  t_rew = 1 - np.tanh(t_tanh_mult * np.abs(t))
  return t_rew

... (other normalized metric functions definitions here)
```

### C.2 Generated Rewards

We show examples of the generated rewards below:

```python
# Pick Reward
Pick_Sparse_Reward = {
    'Is_grasped': 1.0
}
Pick_Dense_Reward = {
  "dis_to_obj": 0.4,
  "xy_dis": 0.2,
  "z_dis": 0.2,
}

# Insert Reward
Insert_Sparse_Reward = {
    'In': 1.0,
    'Is_grasped': -1.0
}
Insert_Dense_Reward = {
    "parallel_dis": 0.2,
    "perpendicular_dis": 0.2,
    "orientation_diff": 0.35
}

# Stack Reward
Stack_Sparse_Reward = {
    'On': 1.0,
    'Is_grasped': -1.0
}
Stack_Dense_Reward = {
    "parallel_dis": 0.35,
    "perpendicular_dis": 0.35
}

# Place Reward
Place_Sparse_Reward = {
    'In': 1.0,
    'Is_grasped': -1.0
}
Place_Dense_Reward = {
    "parallel_dis": 0.35,
    "perpendicular_dis": 0.35
}
```

## D   General Compatibility with Alternative LLMs

To assess the compatibility of our framework with other open-source language models, we evaluate four state-of-the-art foundation models—GPT-5.2, DeepSeek-V3 Liu et al. (2024a), Qwen3-VL-235B Bai et al. (2025), and Llama-3.3-70B Grattafiori et al. (2024)—on isolated task planning and skill learning settings.

For task planning and operator generation, we evaluate the framework across five aspects: (1) the success rate of generating skill operators with correct preconditions and effects, (2) plan possibility (the completeness guarantee that $A^*$ will find a plan whenever one exists under the generated operators), (3) the overall task success rate per model, (4) the average number of attempts, and (5) the zero-shot success rate of reward generation utilizing our **Full MF** formulation. For operator generation (aspects 1–4), each variant runs until 100 correct plans are generated. For reward generation and skill learning (aspect 5), each skill is evaluated over 25 independent trials.

Results for operator and reward generation are shown in Tab. 5 and Tab. 6, respectively. We observe that all evaluated models achieve highly competitive performance, demonstrating the broad compatibility of our framework. Regardless of the underlying architecture, the models successfully generate valid skills with high initial success rates (ranging from 79.37% to 98.04%) and require very few replanning attempts (between 1.02 and 1.26 on average). Crucially, the $A^*$ planner provides a formal guarantee of the possibility of finding an optimal task plan if the required operators are generated, thereby ensuring a 100% overall task success rate across all models via our replanning strategy. Concurrently, in the reward generation phase,

the framework demonstrates exceptional compatibility, maintaining consistently high zero-shot success rates (averaging above 97%) across all evaluated LLMs without any model-specific fine-tuning.

Furthermore, these results suggest a clear scaling trend in our framework: as foundational LLMs improve their reasoning, the initial skill generation success rate increases, which in turn reduces the number of replanning attempts needed. This makes our framework robust to future model upgrades—as base models get stronger, our system's efficiency and performance improve accordingly.

Table 5: **Comparisons on Alternative LLMs for Task Planning.** We evaluate our **LLM+A**$^*$ framework across four state-of-the-art models. All models achieve a 100% end-to-end success rate via replanning, as A$^*$ guarantees plan possibility under the generated operators. Moreover, stronger base models require fewer attempts on average, indicating favorable scaling behavior.

| LLM+A$^*$ Framework | GPT-5.2 | Llama-3.3-70B | Qwen3-VL-235B | DeepSeek-V3 |
|---|---|---|---|---|
| Skill Generation Success Rate (↑) | 79.37% | 90.09% | 92.59% | 98.04% |
| Plan Possibility / Task Planning Success (↑) | **100.0%** | **100.0%** | **100.0%** | **100.0%** |
| Full Design Success Rate (↑) | **100.0%** | **100.0%** | **100.0%** | **100.0%** |
| Average Attempts (↓) | 1.26 | 1.11 | 1.08 | **1.02** |

Table 6: **Comparisons on Alternative LLMs for Reward Generation.** We demonstrate the general compatibility of our **Full MF** approach across different leading foundation models on the "`StowHammer`" task. Results represent the success rate over 25 trials.

| Model | Average | Pick | Place | Open | Close |
|---|---|---|---|---|---|
| Llama-3.3-70B-Instruct | 97.0% | 96.0% | 92.0% | **100.0%** | **100.0%** |
| DeepSeek-V3 | **100.0%** | **100.0%** | **100.0%** | **100.0%** | **100.0%** |
| Qwen-3-235B-Instruct | 99.0% | **100.0%** | 96.0% | **100.0%** | **100.0%** |
| GPT-5.2 | 98.0% | **100.0%** | **100.0%** | 92.0% | **100.0%** |

# E  Comparisons on LLM-Based Skill-Planning Variants

We compare different implementation choices for LLM-based skill-planning systems, including LLM+IL (Behavior Cloning) and TAMP+IL variants that follows common design patterns Ahn et al. (2022b); Mandlekar et al. (2023a).

The LLM+IL baseline utilizes our LLM-based task planner to decompose tasks into symbolic skills, and trains language-conditioned diffusion policies for each skill. Skill execution is composed using predicate-based termination and success checking, conceptually similar to SayCan-style skill composition Ahn et al. (2022b).

The TAMP+IL baseline replaces our RL-trained skills with behavior-cloned policies, following HITL-TAMP-style designs Mandlekar et al. (2023a) in which TAMP provides structured guidance and imitation learning refines local control.

We train separate language-conditioned IL policies for each skill. For example, for the `pick` skill, different object instances are distinguished via CLIP-encoded language tokens such as "pick peg 1". Training data is collected by rolling out expert policies that successfully complete the task. For LLM+IL, we collect 100 trajectories per skill; for TAMP+IL, we collect 50 trajectories per skill, following the training data format in HITL-TAMP.

Results are presented in Tab. 7. IL and RL perform comparably on simpler low-contact tasks such as "`StackAtTarget`". However, for contact-rich tasks like "`PegInHole`", RL significantly outperforms IL-based variants. Furthermore, in unseen task scenarios (e.g., Goal 2), IL methods suffer from limited generalization due to restricted data coverage and the inability of language-conditioned policies to extrapolate to unseen

language tokens. In contrast, our method (LG-SAIL) demonstrates substantially stronger generalization under distribution shift. Moreover, when combined with our automatic domain and reward construction, the RL pipeline enables end-to-end automatic learning and continual adaptation without a human in the loop (i.e., no iterative demonstration collection or manual reward tuning).

Table 7: **Comparisons on LLM-Based Skill-Planning Variants.** We show success rates of different variants on three tasks.

| Task | LG-SAIL | TAMP + IL | LLM + IL |
|---|---|---|---|
| "StackAtTarget" | 1.00 | 0.95 | 0.80 |
| "PegInHole" | 0.95 | 0.70 | 0.50 |
| "PegInHole" (Goal 2) | 0.90 | 0.40 | 0.10 |

## F   More Simulation Results

We present snapshots of simulation rollouts in Fig. 9, Fig. 10, and Fig. 11; additional video results are available on our project website: `https://sites.google.com/view/continuallearning`. Fig. 9 illustrates "`LIBERO-Spatial`", where each row corresponds to a distinct task instance with the target bowl and target plate randomly placed. Fig. 10 presents "`LIBERO-Object`", where each row depicts a task instance with a different target object; the goal in each case is to grasp the target object and place it into the basket. Fig. 11 shows key frames from the execution of "`StackAtTarget`", "`StowHammer`", "`PegInHole`", and "`MakeCoffee`".

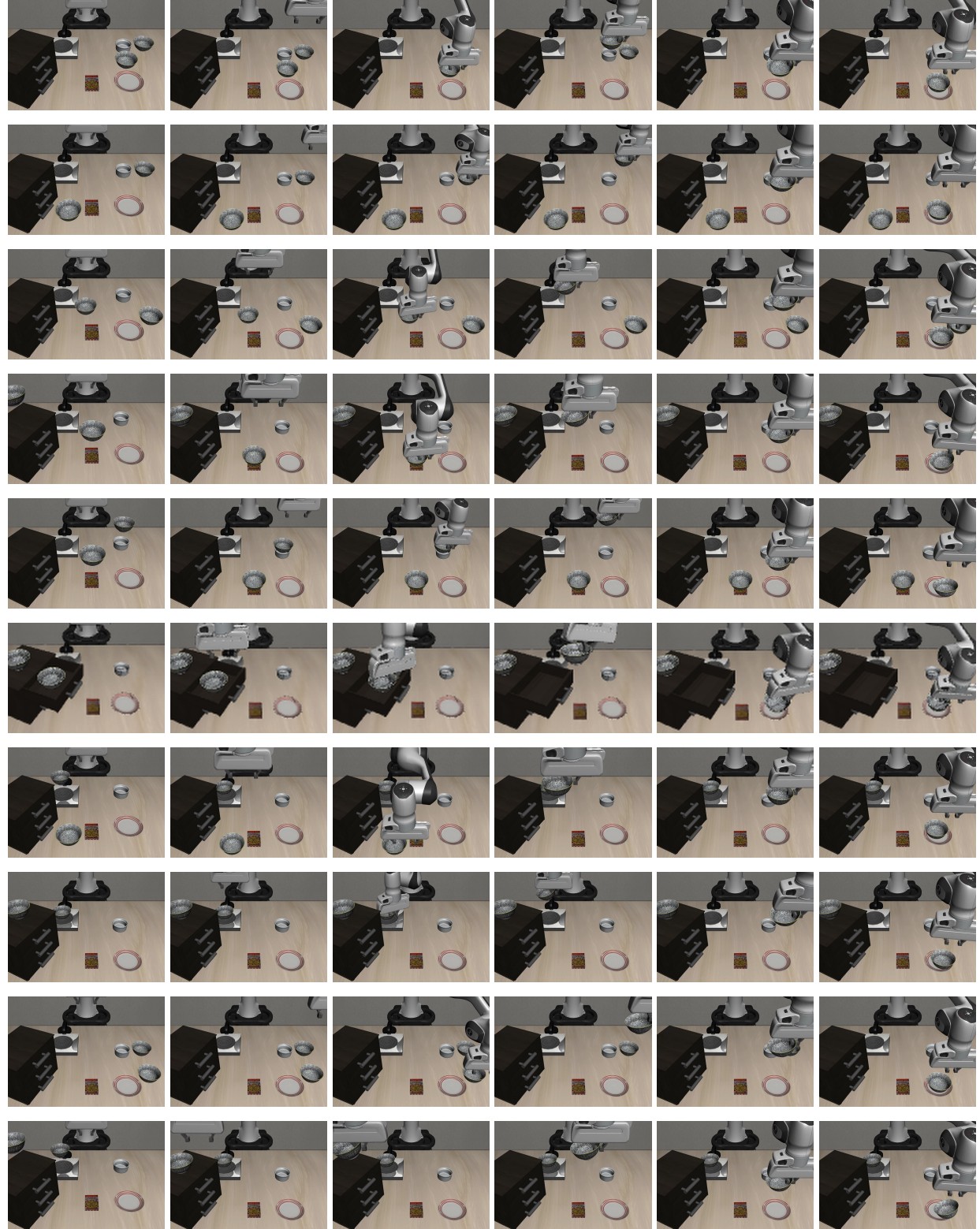

Figure 9: **Task Executions.** We visualize tasks in "`LIBERO-Spatial`" domains.

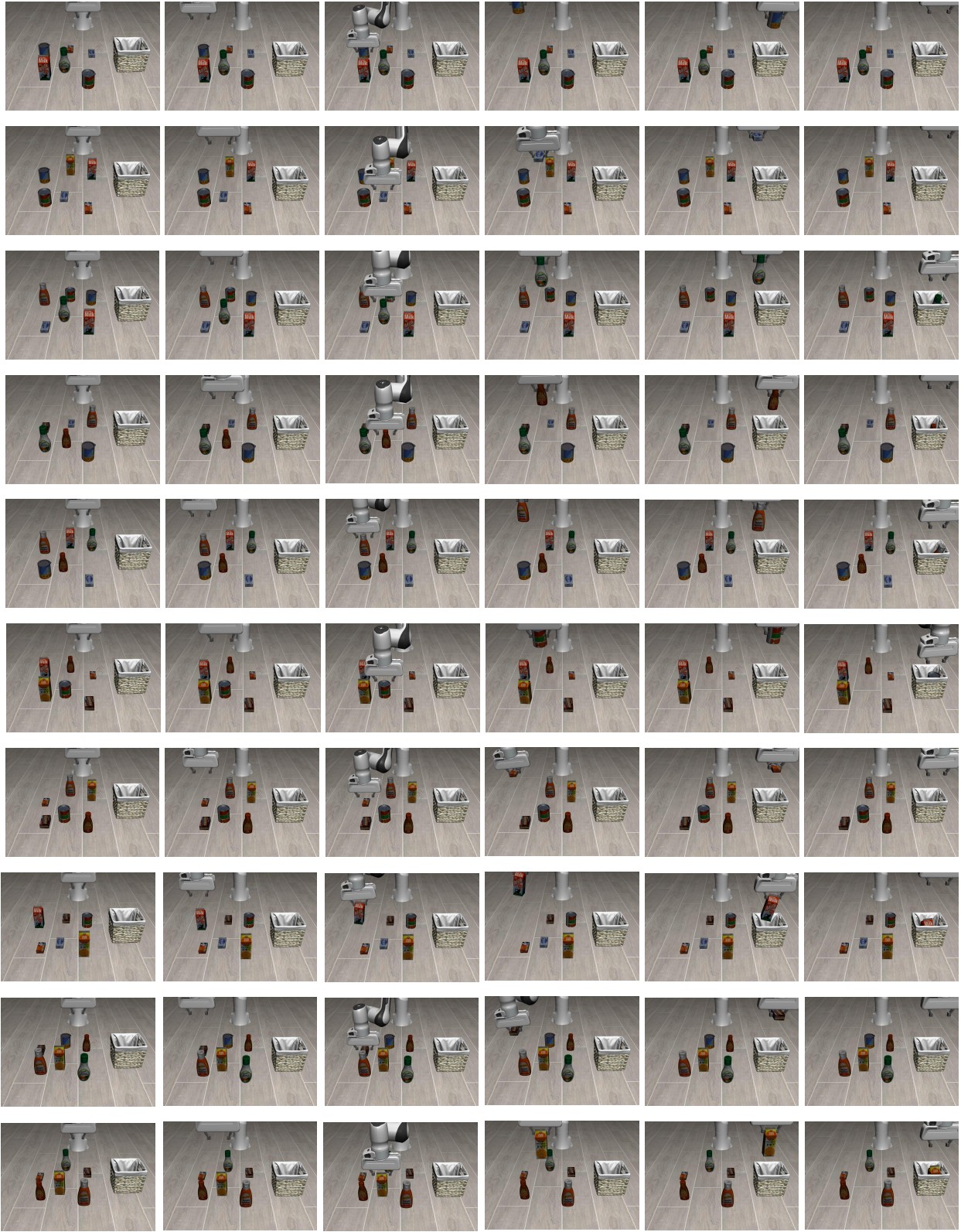

Figure 10: **Task Executions.** We visualize tasks in "`LIBERO-Object`" domains.

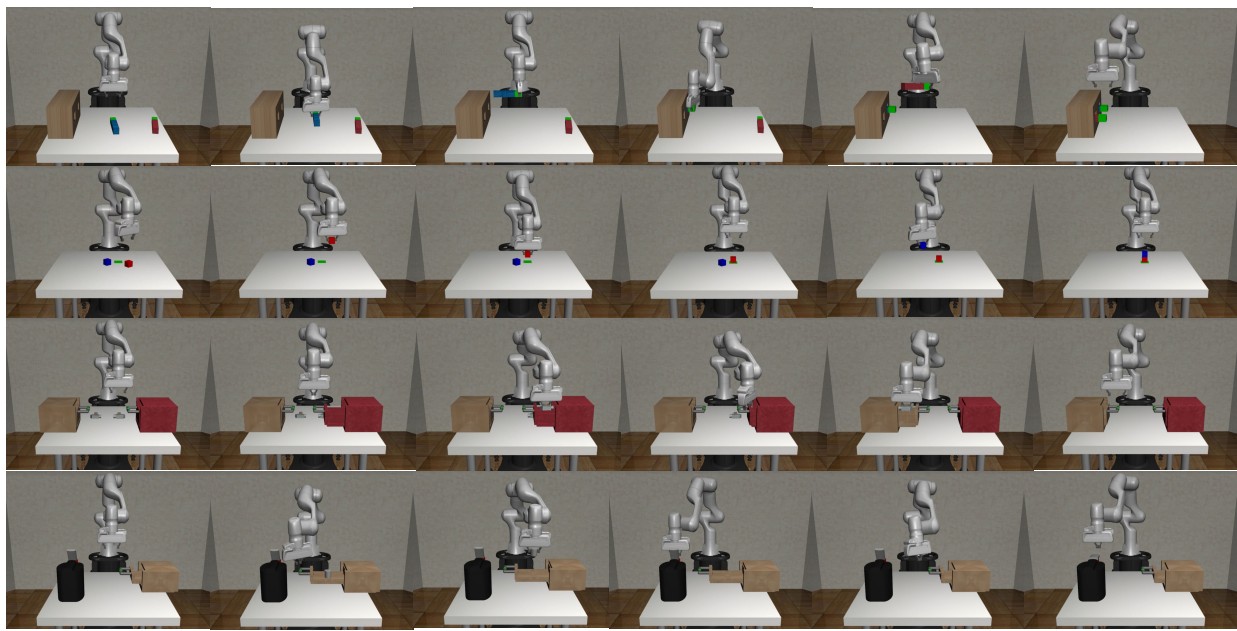

Figure 11: **Task Executions.** We visualize the task execution process of "`PegInHole`", "`StackAtTarget`", "`StowHammer`", and "`MakeCoffee`", respectively.

