# OpenReview forum: "Continual Robot Learning via Language-Guided Skill Acquisition"
_TMLR — Accepted by TMLR_

### Review · Reviewer_dEU1 · 2025-12-20

**Summary Of Contributions:**

This paper gives the hypothesis that Large Language Models can integrate TAMP and DRL for continuous skill learning in long-horizon tasks. It design a LLM-in-the-loop method that using LLM to do task decomposition, reward generation and new skill creation, which can address continuous skill learning without human expert specific knowledge.

**Audience:**

Yes

**Audience Explanation:**

This paper is about robot learning, which is a popular and worth-exploring application in machine learning.

**Broader Impact Concerns:**

No Ethical Concerns.

**Claims And Evidence:**

Yes

**Claims Explanation:**

This paper proposes that previous DRL methods cannot solve long-horizon manipulation problems, and previous TAMP methods need human tailored skills. And it proposes a LLM-in-the-loop method LG-SAIL to solve these problems. And experiences also effectively verify the claim by comparing the method with some methods within DRL and TAMP domains.

**Requested Changes:**

1. I think using LLM is not an efficient way. Giving time comparison with previous methods in DRL and TAMP are also needed.
2. Recently, imitation learning is also popular in robot learning. I think performance comparison with previous imitation learning methods are also needed, for example, iManip [1].

[1] Zheng Z, Cai J F, Wu X M, et al. imanip: Skill-incremental learning for robotic manipulation[C]//Proceedings of the IEEE/CVF International Conference on Computer Vision. 2025: 13890-13900.

---

> ### Author Response · Authors · 2026-02-11
> **Rebuttal by Authors**
>
> Thank you for your time and constructive feedback. We appreciate the recognition that our claims are supported by clear evidence, validated against representative baselines, and of broad interest to the community. Below, we address the raised concerns in detail:
> - **Justification on utilization of LLMs:**
> Our use of LLMs is not to replace efficient search/planning or to be invoked continuously during control. Instead, it targets a key scalability bottleneck in classical TAMP: the need to manually engineer planning domains and low-level controllers, which typically requires expert intervention when transferring to new tasks or domains. In contrast, while DRL can learn policies via trial-and-error, it often relies on carefully shaped dense rewards and task-specific tuning to learn long-horizon behaviors. These practical bottlenecks motivate the recent trend of using LLMs as well as our work to automate task decomposition, planning-domain construction, and reward generation, thereby reducing human effort [1, 2, 3, 4].
> - **Comparison with imitation learning variants:**
> We thank the reviewer for the suggestion and will expand the related work to cover recent imitation-learning approaches, including iManip [5]. Because the original iManip implementation is not publicly available, we include a strong IL baseline in our framework by following common LLM+IL design patterns [6] (LLM-generated task plan with imitation-trained skills) and report the results in Appendix section E. Overall, the IL baseline underperforms our RL-based pipeline: RL-learned skills are more robust to perturbations and contact-rich dynamics, whereas IL is more sensitive to demonstration coverage. Moreover, combined with our automatic domain/reward construction, the RL pipeline supports end-to-end automatic learning and continual adaptation without a human in the loop (no iterative demo collection or manual reward tuning).
>
>
> [1] Eureka: Human-Level Reward Design via Coding Large Language Models, ICLR’24.
>
> [2] DrEureka: Language Model Guided Sim-To-Real Transfer, RSS’24.
>
> [3] Text2Motion: From Natural Language Instructions to Feasible Plans, Autonomous Robots’23.
>
> [4] Generalized Planning in PDDL Domains with Pretrained Large Language Models, AAAI’24.
>
> [5] iManip: Skill-Incremental Learning for Robotic Manipulation, ICCV’25.
>
> [6] Do As I Can, Not As I Say: Grounding Language in Robotic Affordances, CoRL’23.

---

### Review · Reviewer_zS1c · 2025-12-22

**Summary Of Contributions:**

This paper proposes LG-SAIL, a framework that integrates LLMs, Task and Motion Planning, and deep reinforcement learning to support continual skill acquisition for long-horizon robotic manipulation tasks. The core idea is to leverage LLMs for (i) automatic task decomposition, (ii) symbolic operator generation, and (iii) dense reward construction, while grounding and regularizing these outputs using structured predicate and metric-function representations from TAMP. Learned skills are stored in a semantic skill library and reused via embedding-based retrieval to accelerate learning in new tasks and domains.

The paper demonstrates the framework across multiple simulated benchmarks (LEAGUE tasks and LIBERO) and includes real-world deployment on a Franka arm. Empirically, LG-SAIL shows improved learning efficiency over strong baselines, better generalization to new goals, and effective skill reuse across tasks.

Strengths include a well-motivated integration of LLMs with symbolic planning, a carefully designed reward-generation mechanism constrained by metric functions, and a relatively thorough experimental evaluation including ablations and real-world tests.
Weaknesses mainly concern assumptions on predicate availability, reliance on proprietary LLMs, and limited analysis of failure modes and scalability beyond the tested domains.

**Additional Comments:**

Overall, this is a thoughtful and technically solid paper that tackles a genuinely hard problem in long-horizon and continual robot learning. While some aspects—particularly the assumptions on symbolic structure and LLM dependence—limit the scope of generality, the integration itself is non-trivial and well executed. With clearer discussion of assumptions and limitations, this work would make a meaningful contribution to the literature and fits well within TMLR’s emphasis on careful empirical and methodological advances.

**Audience:**

Yes

**Audience Explanation:**

This work is likely to be of strong interest to a subset of the TMLR audience working at the intersection of reinforcement learning, robotics, planning, and language models. In particular, researchers interested in long-horizon robotic manipulation, continual learning, and neuro-symbolic methods will find the framework and empirical insights valuable.

Beyond robotics, the paper also contributes to a broader methodological discussion on how to **constrain and ground LLM outputs** using structured representations, which is relevant to ongoing debates in machine learning about reliability, hallucination, and hybrid symbolic–neural systems. While the paper is application-driven, the design principles—LLM regularization via symbolic structure and metric-function interfaces—are general enough to inspire work in adjacent domains.

**Broader Impact Concerns:**

The work does not raise significant ethical concerns beyond standard issues in robotics and autonomous systems. The primary broader impact considerations relate to deployment safety and reliability, especially when LLM-generated plans or rewards are used in real-world robotic systems. While the current experiments are controlled and limited, a brief discussion on safeguards, verification, or human oversight in more open environments would be appropriate. No major red flags are apparent.

**Claims And Evidence:**

Yes

**Claims Explanation:**

The main claims of the paper are well supported by the presented empirical and methodological evidence. The authors clearly articulate the limitations of prior approaches that either rely on hand-crafted planning domains or use unconstrained LLM outputs, and the proposed design choices are justified through both conceptual arguments and experiments.

The quantitative comparisons against RL, curriculum RL, HRL, and LEAGUE demonstrate consistent gains in task progress across several long-horizon manipulation tasks. The ablation studies, particularly those isolating the role of A* verification in planning and structured metric functions in reward generation, directly support the claims about reduced hallucination and improved robustness. The continual learning experiments on LIBERO further substantiate the claim that the skill library enables faster adaptation over time.

Some claims, especially regarding generality and real-world applicability, are supported by relatively narrow evidence. The real-world experiments are promising but limited in scope, and the reliance on pre-defined predicates means that “automation” still assumes a non-trivial amount of structured prior knowledge. These points do not invalidate the claims but slightly temper their generality.

**Requested Changes:**

Below I distinguish **critical** changes from **non-critical but strengthening** suggestions.

### Critical / Important
1. **Clarify assumptions on predicate and metric availability.**
   The framework assumes a predefined set of predicates and metric functions. A clearer discussion is needed on how much domain engineering this requires in practice, and how sensitive performance is to predicate design quality.

2. **Reduce reliance on proprietary LLMs or discuss alternatives.**
   The use of GPT-4 and OpenAI embeddings is understandable, but the paper should more explicitly discuss reproducibility and expected performance with open-source LLMs, or at least acknowledge potential variability.

3. **Failure mode analysis.**
   While success rates are reported, a more systematic analysis of common failure cases (e.g., incorrect operator effects, reward misalignment, planner–policy mismatch) would strengthen the credibility of the approach.

### Non-critical / Strengthening
4. **Scalability discussion.**
   Some discussion or preliminary evidence on how the approach scales with larger predicate sets, longer plans, or more diverse objects would be valuable.

5. **Broader comparison to recent LLM-planning hybrids.**
   The related work is solid, but a more explicit comparison of conceptual differences with recent LLM-as-planner or code-as-policy approaches could help position the contribution more sharply.

6. **Clarify computational overhead.**
   Reporting approximate LLM query counts and their relative cost compared to RL training would help readers assess practicality.

---

> ### Author Response · Authors · 2026-02-11
> **Rebuttal by Authors**
>
> Thank you for your time and thoughtful feedback. We appreciate the positive recognition of our work’s clear motivation and contributions, its relevance at the intersection of robotics and language models, and its potential to inform and benefit broader research on constraining and grounding LLM outputs with structured representations. We also value the acknowledgement of our sound design choices, technical rigor, and the thorough experimental evaluation supporting our claims. Below, we address the raised concerns in detail:
>
> - **Assumptions on predicate availability:**
> We thank the reviewer for highlighting this point. Predicates and metric functions are lightweight interface assumptions—standard in Task and Motion Planning [1] and LLM–symbolic planning [2,3]---rather than heavy, domain-specific engineering. In practice, our framework only needs a small, reusable set of task-agnostic spatial predicates/metrics (e.g., in, on, grasp, etc.) to describe object–object and robot–object relations. Empirically, we find the method is not sensitive to large predicate libraries: a handful of basic relations already supports diverse long-horizon tasks, since many manipulation problems reduce to achieving target spatial arrangements. Finally, our framework is compatible with predicate discovery and learning methods [4, 5, 6], which can further reduce or eliminate manual predicate design.
> - **General compatibility with alternative LLMs:**
> While we use GPT-4 and OpenAI embeddings in our main experiments, our framework is not tied to any proprietary model. To validate compatibility and expected performance with open-source alternatives, we repeat task decomposition, operator generation, and reward generation using several open-source LLMs and open embeddings; results are reported in Appendix section D. Overall, we observe comparable performance trends, suggesting the framework is generally applicable beyond GPT-4.
> - **Analysis of failure cases:**
> We included additional error analysis in the limitation section.

---

> ### Author Response · Authors · 2026-02-11
> **Rebuttal by Authors (continued)**
>
> - **Scalability w.r.t. problem and task scope (larger predicate sets, longer plans, more diverse objects):**
> We address scalability directly by evaluating a suite of increasingly challenging tasks that (i) require larger and richer predicate sets, (ii) extend planning horizons, and (iii) involve diverse objects and spatial arrangements. For instance, MakeCoffee demands multiple predicates to capture the coffee machine’s operational states; StowHammer requires reasoning over cabinet–object spatial relations and stresses long-horizon execution (8 skills, ~1600 environment steps); and LIBERO tasks test adaptability to novel layouts and diverse object geometries through continual learning. Across these settings, the framework remains effective without re-tuning core components, suggesting graceful degradation as symbolic complexity and task scope increase.
> - **Additional comparison with LLM planning + skill policy variants:**
> We thank the reviewer for the suggestion. We will sharpen the positioning by explicitly contrasting our approach with recent LLM-as-planner hybrids (e.g., integration with BC policies or motion planner [7, 8]). Our key distinction is that the pipeline is fully automatic: the LLM is used offline to generate task decompositions, operators, and reward specifications, and skills are then learned without human-in-the-loop (no manual reward shaping, no teleoperation and iterative demo collection). This design is specifically intended to adapt to new tasks/domains with minimal engineering: when the task changes, the system can regenerate the high-level structures and relearn/compose skills automatically rather than requiring new task-specific demonstrations. We also include an LLM+BC (IL) variant following common design patterns [7] and show it underperforms our RL-based pipeline (see Appendix section E), consistent with RL providing more robust behaviors under perturbations and contact-rich dynamics while BC is sensitive to data coverage.
> - **Computational overhead:**
> We analyze the cost of LLM-based task decomposition and planning in Table 2, reporting the average number of LLM calls per task. Concretely, our method uses a symbolic planner to verify the LLM-generated operators and only re-queries the LLM when the planner fails to reach the goal. This verification-and-regeneration loop both improves planning reliability and reduces LLM usage: it achieves 100% planning success versus 36% for the 1-shot baseline, while requiring fewer attempts on average (1.6 vs. 2.6 in prior LLM-based planning). Importantly, this overhead is incurred only during high-level planning and is typically negligible relative to the cost of RL training and environment interaction.
>
> [1] Learning Neuro-Symbolic Skills for Bilevel Planning, CoRL’22.
>
> [2] Generalized Planning in PDDL Domains with Pretrained Large Language Models, AAAI’24.
>
> [3] PDDLEGO: Iterative Planning in Textual Environments, ACL’24.
>
> [4] SORNet: Spatial Object-Centric Representations for Sequential Manipulation, CoRL’21.
>
> [5] Grounding Predicates through Actions, ICRA’22.
>
> [6] Predicate Invention for Bilevel Planning, AAAI’23.
>
> [7] Do As I Can, Not As I Say: Grounding Language in Robotic Affordances, CoRL’23.
>
> [8] VoxPoser: Composable 3D Value Maps for Robotic Manipulation with Language Models, CoRL’23.

---

### Review · Reviewer_3M37 · 2026-01-28

**Summary Of Contributions:**

This paper presents LG-SAIL, a framework and approach for combining Task and Motion Planning (TAMP) with Deep Reinforcement Learning (DRL) for lifelong robot learning, and using large Language Models to bridge the gap between them. The paper’s main thrust is that both DRL and TAMP have their unique advantages and can complement one another. While this has been shown in some recent work, the paper then introduces LLMs as the third component, which can be used to generate dense rewards, automate task decomposition, and create skills. The pipeline presented in the paper is set up so that each component helps address shortcomings from others: expert domain knowledge needed in TAMP can be replaced by using LLM’s breadth of semantic knowledge, LLM hallucinations can be fixed by grounding using TAMP’s structural information, hand-crafted dense rewards for RL can then be omitted in favor of rewards generated from LLMs. The paper also includes the use of a skill library, which allows previously learned skills to be reused to warm-start new skills. The paper benchmarks LG-SAIL against previous methods, including LEAGUE, Soft-Actor Critic, Curriculum RL, and Hierarchical RL on three domains from the LEAGUE paper; shows generalization to new goals and environments by changing aspects of the LEAGUE domains and by using tasks from the LIBERO benchmark; and performs ablations on its use of LLMs for task planning and reward generation. Finally, the paper also demonstrates that its results hold in the real world by applying LG-SAIL to two real-world tasks.

Key strengths of the paper include the description of its pipeline, the emphasis on how each component strengthens the overall approach, and the breadth of the experimental design and ablations. A primary weakness of the paper is the omission of certain tasks in some experiments, without clarification of why those choices were made.

**Additional Comments:**

On a minor note, the paper contains a few grammatical or formatting errors that need correction and would benefit from general proofreading by the authors. For example, in Figure 9, rows 5 and 6 show the same rollout.

**Audience:**

Yes

**Audience Explanation:**

The paper's approach will be of interest to researchers and audience members working in the fields of robot learning, motion planning, Large Language Models (LLMs) for robotics, and reinforcement learning (RL).

**Broader Impact Concerns:**

The paper has no Broader Impact statement section. No broader, unique impact concern comes to mind having gone through the paper.

**Claims And Evidence:**

Yes

**Claims Explanation:**

The paper robustly and clearly makes its case for why integrating LLMs with DRL and TAMP  is needed, supported by motivation, evaluations, and a literature review, and adequately presents the various components of LG-SAIL.

**Requested Changes:**

1. The authors need to clarify why certain tasks are chosen for certain experiments. For example, in the baseline comparison(Section 5.2 and Figure 4), why is MakeCoffee the only task of the original four from LEAGUE that is omitted?

2. Similarly, in the generalization tests on LIBERO-Object and LIBERO-SPATIAL, it would have been useful to compare LG-SAIL against LEAGUE at the very least. The authors should include this or provide sufficient justification for omitting them.

---

> ### Author Response · Authors · 2026-02-11
> **Rebuttal by Authors**
>
> Thank you for your time and constructive feedback. We appreciate the positive assessment of our work’s clear technical and implementation details, the coherent integrated system design, and the comprehensive experimental evaluation supporting our claims. Below, we respond to each concern in detail:
>
> - **Task design rationale and additional baseline results:**
> In Sec. 5.2 and Fig. 4, we focused on tasks that best isolate the contribution of skill reuse across task domains under minimal changes to the low-level skill library. MakeCoffee is included for exactly this reason—it requires composing and transferring reusable object-centric skills in a semantically different domain rather than only reordering similar manipulation primitives. To address the reviewer’s concern, we have added the LEAGUE baseline on MakeCoffee in the revised version.
> For the LIBERO-Object and LIBERO-SPATIAL generalization tests, our goal is to stress adaptation to novel object shapes and spatial arrangements at scale. Although LEAGUE can attain comparable performance when the planning domain (operators/predicates) and skill rewards are manually specified, our method differs by automating planning-domain construction and reward generation via LLMs.
>
> - **Grammar, typos, and formatting:**
> We have carefully proofread the manuscript and addressed all identified issues in the revised version.

---

> > ### Comment · Reviewer_3M37 · 2026-02-26
> >
> > I thank the authors for their responses to my comments.
> >
> > I think the paper is in a much stronger position now.

---

### Author Response · Authors · 2026-02-25
**Follow-up on our response to reviewer comments**

We’re writing to kindly follow up on our submitted response to the reviewer comments for our manuscript. Please let us know if any additional clarification or information would be helpful. Thank you for your time and consideration.

---

### Decision · Action_Editor_N89k · 2026-02-26

**Recommendation:** Accept with minor revision

**Audience:**

Yes

**Audience Explanation:**

All reviewers responded positively. The problem addressed in this work is likely to attract broad interest from researchers working on LLMs for robot learning.

**Claims And Evidence:**

Yes

**Claims Explanation:**

All reviewers responded positively to this question and unanimously agreed that the key contributions of the submission are well supported by the empirical and methodological evidence.

**Resubmission Of Major Revision:**

The authors may consider submitting a major revision at a later time.